# Morphological Characteristics of Catalyst Layer Defects in Catalyst-Coated Membranes in PEM Fuel Cells

**Muneendra Prasad Arcot [1,*], Magnus Cronin [1], Michael Fowler [1,2] and Mark Pritzker [1,*]**

[1] Department of Chemical Engineering, University of Waterloo, 200 University Avenue West, Waterloo, ON N2L3G1, Canada
[2] Green Energy and Fuel Cell Research Group, University of Waterloo, 200 University Avenue West, Waterloo, ON N2L3G1, Canada
* Correspondence: muneendraprasad@gmail.com (M.P.A.); pritzker@uwaterloo.ca (M.P.); Tel.: +1-548-333-4117 (M.P.A.)

**Abstract:** Catalyst layer defects and irregularities in catalyst-coated membrane (CCM) electrodes affect the lifetime of polymer electrolyte membrane fuel cells (PEMFCs) during their operation. Thus, catalyst layer defects are important concerns for fuel cell manufacturers and prompt the development of quality control systems with the aim of fabricating defect-free electrodes. Consequently, the objective of this study is to gain a fundamental understanding of the morphological changes of real catalyst layer defects that have developed during CCM production. In this paper, missing catalyst layer defects (MCLD) formed during the decal transfer process are investigated through a nondestructive method using reflected light microscopy. The geometric features of the defects are quantified, and their growth is measured at regular time intervals from beginning-of-life (BOL) to end-of-life (EOL) until the OCV has dropped by 20% of its initial value as per a DOE-designed protocol. Overall, two types of degradation are observed: surface degradation caused by catalyst erosion and crack degradation caused by membrane mechanical deformation. Furthermore, catalyst layer defects formed during the decal transfer process were found to exhibit a higher growth rate at middle-of-life (MOL-1) and stabilize by EOL. This type of study will provide manufacturers with baseline information to allow them to select and reject CCMs, ultimately increasing the lifetime of fuel cell stacks.

**Keywords:** catalyst layer; defects; pinholes; catalyst-coated membrane; MEA; PEMFC

## 1. Introduction

PEMFCs have emerged as promising, eco-friendly devices with numerous applications in the automotive industry. The use of PEMFCs is aimed at resolving two critical problems associated with conventional fossil fuel combustion vehicles: the contribution of vehicle emissions to climate change and petroleum reserve depletion.[1] Many automotive manufacturers such as Hyundai, Toyota, Honda, and Ford have introduced fuel cell vehicle (FCV) technology aimed at harnessing its potentially high efficiency, reliability, continuous mode of operation, and zero emissions [2,3]. Greater interest in fuel cell systems has spurred significant growth in this sector. In 2021 alone, the number of fuel cell system shipments from manufacturers to end-users increased by 75.7% relative to the previous year across the world. The total power generated by these fuel cells amounts to 2330 MW [4]. The manufacturing of reliable and effective fuel cell components demands the development of a comprehensive quality control system to identify material defects [5]. As such, the US Department of Energy (DOE) has introduced a fuel cell program to work in collaboration with manufacturers, universities, and national laboratories to produce high-quality fuel cell products for use in fuel cell stacks [6]. In 2017, the DOE released data showing that the percentage of electrodes that fail due to imperfections is 2.5%

during catalyst production, 2.5% during catalyst coating, 3.0% during decal transfer, 3.0% during die cutting, and 0.5% during hot pressing [7].

The basic element of a fuel cell is the membrane electrode assembly (MEA). The MEA is composed of a catalyst-coated membrane (CCM) hot-pressed to gas diffusion layers (GDLs) on either side [8]. Each CCM consists of a polymer electrolyte membrane coated on one side with the anode catalyst layer (ACL) and the other side with the cathode catalyst layer (CCL). CCM-based MEAs have many advantages compared to conventional GDE-based MEAs (gas diffusion electrodes with a catalyst coated on GDLs) such as lower contact resistance between the catalyst layer and electrolyte membrane, easier gas transport, more effective catalyst utilization, and thinner electrodes. The CCM is the key component of a PEMFC since it contains the expensive platinum electrocatalyst, which must provide a high-active surface area on the order of 70–120 $m^2g^{-1}$ to achieve the desired results [9,10]. The development of high surface area catalysts has helped reduce the amount of platinum required per unit area and reduce the thickness of the catalyst layer from 5 μm–50 μm to between 0.2 μm–10 μm [11]. Several studies have demonstrated the advantages of a thinner catalyst layer in lowering the electron and proton transport resistances, increasing the $O_2$ concentrations within the cathode, reducing the amount of platinum usage, and minimising the overall system cost [12–14]. In addition, thin catalyst layers are beneficial for the three-phase catalyst reaction, gas permeability, fluid transport, electrical conductivity, and ionic conductivity [15,16]. If any of these properties are adversely affected, fuel cell performance suffers significantly. Research has shown that thin catalyst layers are advantageous since they improve the kinetics of the oxygen reduction reaction, minimize catalyst loading, reduce electrode resistance, and increase current density [17].

Any damage to the fuel cell components during the manufacturing process ultimately leads to the formation of defects. In particular, defects on the catalyst layer can dramatically affect fuel cell performance, cost, and stability [18–20]. Unnecessary time and money are spent tearing apart fuel cell stacks to remove a single faulty cell. Importantly, it is critical that defects developed during fuel cell production be examined and characterized to differentiate between those that are minor and those that are fatal. Perhaps the most important requirement for the mass production of fuel cell components is to improve the quality control inspections aimed at identifying CCM defects and predicting CCM lifetime [21,22]. Better quality control inspection would help reduce CCM imperfections that stem from errors associated with, for example, catalyst ink preparation and catalyst coating methodology [23–25] and variations in the thicknesses of the catalyst layers and electrolyte membranes [26,27].

Understanding the various types of defects, their origins during manufacturing, and their impact on cell performance is extremely important in developing a quality control process. If this information is combined with defect detection guidelines developed by electrode manufacturers, material suppliers, production engineers, research laboratories, and governments, a systematic approach to quantifying defects can be developed. In the future, different parties should provide their own perspective on classifying defect severity and priority to eventually formulate a consistent decision-making process.

To achieve an accurate CCM quality control inspection system, this research focuses on real catalyst layer defects. In collaboration with industrial partners, this study focuses on the examination of the orientations and irregularities of CCM catalyst layer defects and missing catalyst layer defects (MCLD) that have been introduced during MEA fabrication. The objective of this work is to visually inspect the morphological changes of these catalyst layer defects as they propagate throughout the aging process by implementing a nondestructive investigation method. The defective CCMs/MEAs are then electrochemically tested in a stack to relate these defects to fuel cell performance in realistic environments. Key concerns of this study are the factors that lead to defect propagation and ultimately trigger catalyst layer failure. Critical evidence will be provided on morphological changes of the catalyst layer defects during the aging process that leads to performance losses in

CCM electrodes, which can be used to guide future research. The data obtained in this study will provide important information for fuel cell electrode manufacturers and researchers on the evolution of catalyst layer flaws over time. Such information can be used to improve the design and optimization of thinner catalyst layer structures. This research will be followed by future work on the effect of defect degradation on cell performance under typical operating conditions including steady-state OCV and cyclic OCV.

## 2. Experimental

### 2.1. Scope of Defect Analysis

One of the major challenges facing CCM electrode manufacturers is associated with catalyst layer defects developed during fabrication. These defects can range from micrometers to millimeters in size and have very irregular geometry, which can have a strong impact on degradation. Another difficulty in controlling MEA fabrication is that some of these defects have a significant effect on fuel cell performance, while others do not. Unfortunately, it is usually difficult to predict *a priori* what category any given defect will fall into after an MEA is formed during hot pressing. Many researchers have examined the effect on overall cell performance of artificial defects occurring at random electrode locations. However, a research gap still exists in analyzing more realistic defects and their evolution during aging. In this study, the morphology of realistic catalyst layer defects generated in commercial CCM production lines is analyzed at various stages of aging.

### 2.2. CCM Defect Analysis Framework

Specific commercial CCMs with an active area of 48 cm$^2$ and catalyst layer defects developed during fabrication in a commercial production line were obtained from an industrial fuel cell manufacturing partner [15,26]. The catalyst layers were made by mixing a Pt/C catalyst diluted with deionized water and ethanol mixed with an ionomer binder solution. The catalyst ink was spray-coated onto a decal substrate and dried at 80 °C. The catalyst layer on the decal substrate was then transferred onto a PFSA Nafion membrane using the decal transfer method in a hot press at 140 °C for three minutes. The thickness of the cathode catalyst layer was measured to vary between 10 μm and 15 μm. During decal peeling, tiny portions of catalyst can remain on the decal substrate, forming an irregularly shaped defect on the catalyst layer. These defects are defined as a missing catalyst layer or a completely removed catalyst layer. The effectiveness of decal transfer can be characterized in terms of the transfer ratio which is defined as the percentage of the total area of a fresh unused CCM surface that is defect free. The transfer ratios of the catalyst layers on the four defective CCM electrodes investigated in the study were found to be ~97% ± 2%.

Figure 1 presents a novel framework comprising four steps developed to investigate real catalyst layer defects, which should benefit the quality control procedures used by fuel cell manufacturers. In step one, the catalyst layer defects in CCMs were thoroughly examined under the optical microscope to study their morphological characteristics, including defective area, length, width, and aspect ratios. IR thermography was used in step two to identify defect criticalities such as thinner areas or pinholes. In the event that any pinholes were found during this stage of the inspection, the CCM was removed from the batch and then proceeded to step four where it was subjected to electrochemical analysis (impedance and polarization measurements) for future reference. As a CCM ages, a few defects in the electrode may propagate to develop pinholes. Such defective areas facilitate gas crossover through the membrane, causing some short-circuiting and a reduction in the net voltage across the cell. Therefore, we have used the OCV as a measure of the state of the cell and classified aged CCMs as being middle-of-life (MOL) or end-of-life (EOL). More details regarding the criteria defining each of these states are given in Section 2.2.3. When the CCM reached EOL, it proceeded to step four and electrochemical characterization. This framework is novel because it provides manufacturers with a nondestructive

CCM defect analysis tool. More details of this procedure are discussed in the following subsections.

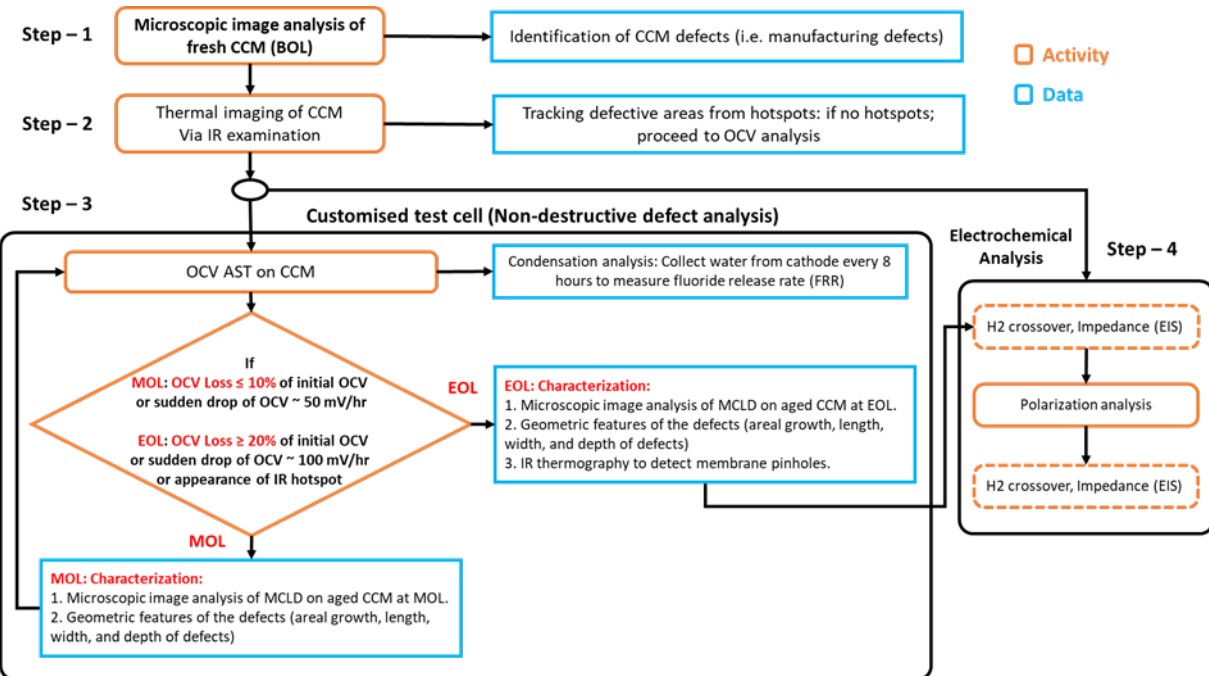

**Figure 1.** Nondestructive CCM defect analysis framework combined with electrochemical analysis.

### 2.2.1. Microscopic Imaging

After batches of CCM electrodes were fabricated on the production line, samples were carefully examined using a Nikon Eclipse MA 200 (Tokyo, Japan) inverted metallurgical reflected light microscope,, in order to identify defects on the catalyst layers. The defective CCMs were separated from nondefective CCMs for further microscopic analysis. Nikon Imaging Solution (NIS) software, NIS-Elements L (Tokyo, Japan) was used to image the surface morphology and generate details on defects such as surface profiles, orientation, dimensions, and aspect ratios [28]. In addition, 3D defect maps were generated using the digital information obtained from Z-profile scanning. The extended depth-of-focus (EDF) module in the NIS software allows users to generate a Z-stack of images in one image by selecting the focused regions from each frame and piecing them together. The depth-coded alpha blending module accumulates all the pixel intensity values of the Z-sequence and finely tunes them into a 3D-rendered image. A gradient is incorporated into the Z-stack planes by coloring the lower, middle, and upper sections differently and enabling users to easily distinguish the position of the defective areas. The software also allows users to manually focus on a defect region in the catalyst layer and produce a final image. This procedure has been used throughout the life of a CCM electrode to characterize the defects at BOL and monitor their evolution at MOL and EOL. Finally, the geometry and size of the defects were characterized with respect to their area, length, width, and aspect ratio.

### 2.2.2. IR Thermography

After the initial investigation of the defects, the CCMs were further examined using IR imaging to confirm the presence of pinholes. IR thermography was conducted by passing hydrogen gas (20% hydrogen diluted with 80% nitrogen) over the anode and exposing the cathode to atmospheric oxygen. The IR camera and the electrode were placed inside a dark environmental chamber to eliminate external light reflections while the camera lens was located 1 m away from the cathode. Any pinholes in the membrane or leak areas facilitate hydrogen crossover from the anode to the cathode. The direct combustion of the

crossover hydrogen with oxygen in the presence of the Pt catalyst generated heat (infrared energy) that appears as a hotspot on the IR image at the location of the pinhole. The software in the camera converted the IR image into a thermal image. The thermography image was displayed by assigning a specific color to each thermal energy level.

2.2.3. Open Circuit Voltage-Accelerated Stress Test

The propagation of defects was monitored while the CCM was subjected to an open-circuit voltage accelerated stress test (OCV-AST). This OCV diagnostic tool was chosen to accelerate the degradation of the electrodes in order to investigate the failure mechanism and predict the lifetime of the electrode [29,30]. An abnormal OCV in the cell indicated electrode failure due to phenomena such as pinholes in the membrane or gasket leaks. DOE-recommended protocols were used for this OCV-AST operation [29,30]. OCV-AST tests were carried out on CCMs to evaluate the chemical deterioration of catalyst layer defects up to the point where the OCV dropped to 20% or less of its initial value. The results obtained for three selected CCMs are reported in this study: CCM-1-MCLD (The details of MCLD is discussed in Section 3.2), CCM-2-MCLD (which has similar types of defects to CCM-1 in terms of shape, size, and orientation), and CCM-baseline (nondefective CCM). Before a CCM was subjected to the OCV-AST, it was examined by microscopic analysis and IR thermography to confirm that no pinholes or leak spots were present at BOL. Only pinhole-free CCMs proceeded to the OCV-AST. A custom-designed test cell was used during these ASTs to enable the user to investigate catalyst layer defects at MOL and EOL. Some details of this test cell are given below. A more extensive description of this test cell is provided in our previous study [31].

1.  The test cell design involved transparent polycarbonate plates on the outside of two sets of lands and channels that were 2 mm wide and 1.5 mm deep [31]. This design allowed the user to monitor thermal changes on the electrode through IR thermography to easily identify leaks or pinholes.
2.  The test cells were operated using gaskets of varying thicknesses (0.5, 1, 3, 5, 6, and 8 mm) without gas flow plate lands or channels to prevent the compression and/or damage of the catalyst surface by lands/channels. The gaskets provided enough space for the membrane to swell without being damaged by the flow paths. A 6 mm thick gasket was found to yield the best results without any external damage to the catalyst layers. Overall, the use of this gasket ensured that defect (catalyst layer cracks) formation was caused solely by mechanical deformation due to membrane swelling in the presence of humidified gases rather than by mechanical deformation due to impinging land/channels.
3.  To provide an optimal electrical contact surface, a small portion of the anode and cathode GDL was extended to the end of the test cell to measure the potential difference (OCV) during the experiments. The OCV experiments were performed without hot pressing the GDL to the CCM. Since the GDL was not compressed and the CCM was not confined by the land/channels, the morphology of defects should not be affected by the GDL fibers and/or flow channel plate indentations. Therefore, the propagation of defects should be driven by the chemical and mechanical degradation caused by the reaction gases.

This testing method for catalyst layer defects was successful in investigating CCMs during MOL while mitigating any external damage. During the OCV analysis, two criteria were set to investigate the evolution of defects:

A.  MOL: Examination of the CCM surface during MOL by optical microscopy was conducted if the OCV dropped 10% or less below its initial value or the OCV dropped suddenly by ~50 mV/h. MOL image analysis included characterization of the geometric features of the defect.
B.  EOL: The experiment was terminated and EOL inspection of the CCM was conducted when the OCV dropped 20% or more below its initial value, the OCV dropped

suddenly by ~100 mV/h, or if IR thermography detected any hotspots on the electrode. After characterization by optical microscopy and IR thermography were carried out, the aged CCM was subjected to electrochemical polarization analysis and $H_2$ crossover measurements.

2.2.4. Electrochemical Analysis

Polarization Analysis

Polarization (I-V) measurements were conducted using a G-50 fuel cell test station with an RBL 232 (TDL electronic device) electronic load box. Hydrogen gas and air were fed to the anode and cathode at stoichiometric ratios of 1.5 and 2, respectively, while the cell was kept at a constant temperature of 90 °C using a water coolant plate. I-V curves were obtained at a temperature of 90 °C with an 80% relative humidity at both the anode and cathode. The cell voltage was measured over the current density range beginning at the maximum of 2A/cm² and decreasing to 0 A/cm² (i.e., OCV). An RH of 80% was found to lead to the optimum fuel cell performance and so the RH was set at this level for the remainder of the study.

Hydrogen Crossover Measurement by LSV and FER

Hydrogen crossover was measured to assess the health of the MEA at different stages. Hydrogen crossover was measured by linear sweep voltammetry (LSV) using a BioLogic Science potentiostat model VMP3 (Seyssinet-Pariset, France) with an HCP-1005/100A booster and EC lab V10.39 software. To measure the hydrogen crossover rate, $H_2$ was supplied to the anode while $N_2$ was fed to the cathode. As the MEA aged, some $H_2$ from the anode side would be expected to cross over through defects and react at the cathode to generate a crossover current. To isolate the effects of hydrogen crossover, the voltage was scanned from 0V to 0.7 V at a scan rate of 2 mV/s. The crossover current was then determined by averaging the current measurements over the potential range from 0.4 V to 0.5 V.

The extent of the leaching of ionomer from the catalyst layer and membrane was determined by measuring the fluoride emission rate (FER) in the cathode water collected at the outlet streams from the cell at regular 10 h time intervals. A Dionex DX 500 ion chromatographic analyzer (Sunnyvale, CA, USA) was used to measure the fluoride ion concentration in the outlet water.

## 3. Results and Discussion

### 3.1. Microscopic Investigation of Catalyst Layer Defects in CCMs

In this study, defects were found only on the cathode and not on the anode layers. The MCLD is one of the most common defect types that occurs during the mass production of CCMs. Figure 2a is a schematic diagram depicting the decal transfer method by which CCMs are fabricated. The red arrows in the image indicate two MCLDs formed during the decal removal process where the catalyst is not completely transferred from the decal substrate to the CCL. Figure 2b shows the reflected light microscope setup used to investigate the catalyst layer defects shown in Figure 2a. The CCM samples are first attached to a plastic frame to dampen any membrane vibration that may occur. To identify catalyst layer defects, a beam of white light with 20% intensity was directed (transmitted light) onto the ACL of the CCM by facing the CCL toward the reflected light microscope camera. Then a motorized stage connected to the x-y planes of the microscope rastered over the CCM sample at a uniform rate so that the complete sample area could be inspected. At nondefective catalyst areas, the thick CCL blocked light from transmitting through the CCM. However, light was transmitted through the CCM (irregular catalyst areas, zero catalyst, cracks, and pinholes) at thinner defective catalyst areas and merged with green reflected light from the objective lens to produce a magenta/pink colour across

the defective areas. The distinct colour marking of the sites allowed specific defects on the CCM to be identified and characterized. Generally, more intense regions of light indicated thinner catalyst layers. Specific regions of interest (ROI) were further investigated using the dark field mode to provide more detail on the MCLDs and cracks in the catalyst layer. Finally, a complete areal inspection was carried out by using Image J software to stitch together a high-resolution digital image from the numerous microscopic video images of the CCMs. The entirety of the 48 cm² stitched CCM at MOL-1 and EOL is shown in Figure 3. A 10% overlap in the area from one image to the next was maintained throughout the stitching procedure to ensure the best correlation between them. The software adjusted the image edges in both directions until the best match of edge features was found. The ridge-like markings appearing in the final images are artefacts of this stitching procedure that arise due to slight differences in the brightness levels of the adjacent images. Most importantly, they have no effect whatsoever on the appearance and features of defects such as cracks and MCLDs.

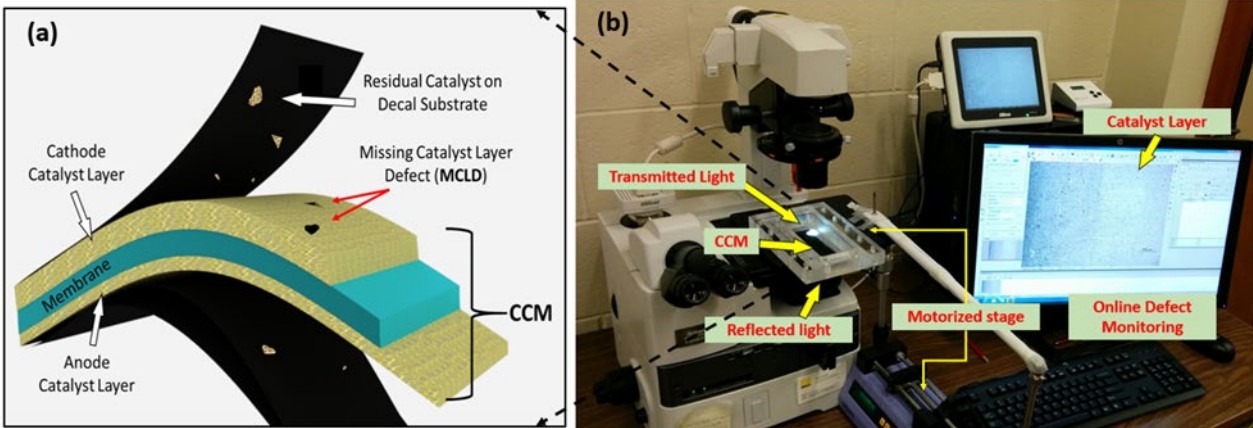

**Figure 2.** (**a**) Fabrication of CCM using decal transfer method and (**b**) reflected microscope setup for investigating catalyst layer defects in CCMs.

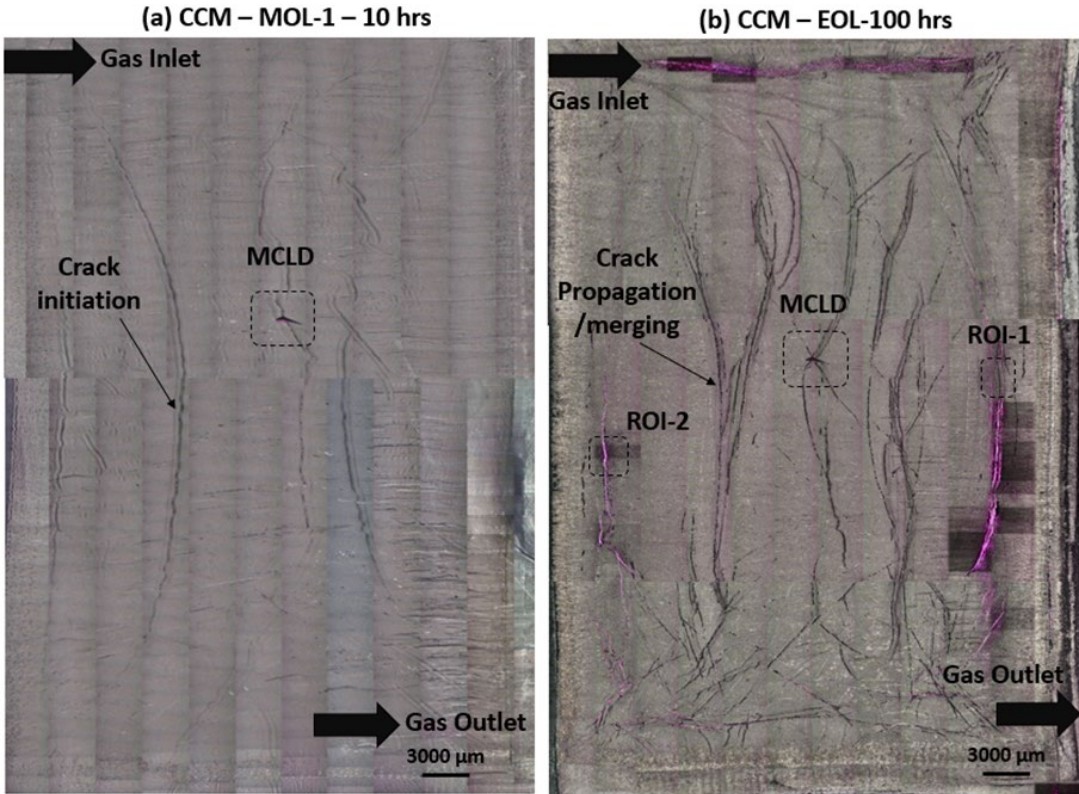

**Figure 3.** (**a**) Stitched optical microscopic image of CCL of defective CCM with MCLD captured after (**a**) 10 h (MOL-1) and (**b**) 100 h of OCV-AST (EOL). Locations ROI-1 and ROI-2 are specifically selected to investigate the delamination of CL.

Once inspected in this way, CCMs to be aged proceeded to the OCV-AST conducted in the custom-designed test cell discussed in Section 2.2.3. The microscopic image analysis followed the procedure described in the flowchart/framework in Figure 1. Figure 3 shows the 360-stitch optical image of an entire CCM aged after 10 h and 100 h of OCV-AST. The gas inlet and outlet directions are indicated at the top left and bottom right, respectively. The image (Figure 3a) obtained after 10 h clearly revealed two types of defects, an MCLD and a few cracks. Although not included here, the image of the CCM obtained at BOL showed the presence of only the MCLD defect prior to aging. The presence of this defect prior to aging was not surprising since this type of defect is usually associated with the decal transfer step during CCM manufacturing. The formation of cracks only after the CCM has been subjected to the OCV-AST was consistent with the expectation that this defect was caused by mechanical deformation driven by a swelling of the membrane. During fuel cell operation, the membrane takes up moisture from the humidified gases with which it comes into contact, causing it to swell and increase in thickness. As the membrane swells, it pulls on the CL layer. However, since this swelling typically occurs nonuniformly within the membrane, this led to gradients in stress over different regions of the CL and crack formation.

Crack growth and propagation were observed in this study to occur during three stages: crack initiation at MOL-1 (10 h), crack propagation at MOL-2 (50 h), and crack propagation/merging/delamination at EOL (100 h). As shown in Figure 3a, the number of cracks observed at MOL-1 was relatively modest. The stitched microscopic image of the aged CCM after 100 h of OCV-AST shown in Figure 3b revealed that many new cracks had formed and that they had propagated significantly by the time EOL was reached. Two regions in the EOL sample denoted as ROI-1 and ROI-2 had been specifically selected for closer examination of the damage to the catalyst layers. ROI-1 corresponded to an area where cracks had severely propagated in the catalyst layer. ROI-2 was an area where delamination had occured as a result of crack propagation caused by membrane swelling.

The depth of the catalyst layer defect in ROI-2 had been determined using Z-profile and 3D imaging. This analysis showed that the catalyst layer in these areas had not been completely removed and that the delaminated or detached material amounted to ~50–70% of the original thickness over an average width of ~50 μm–150 μm, leaving behind thin portions of the catalyst attached to the electrolyte membrane. Furthermore, large catalyst layer pores with an average diameter of ~25 μm were observed in the leftover thin catalyst layers. Further examination of this type of delamination in CCMs and an image of the pores present in the remaining CL after delamination are presented in Section 3.3.2.

### 3.2. Degradation of Catalyst Layer Defect—MCLD

Figure 4 shows an overview of the morphological changes of an MCLD (dark triangular area) originally formed due to improper decal transfer. The MCLD here was the same as the one shown in Figure 3, except that the details of its structure were now clearly visible due to the much higher resolution of the image. The detailed geometric growth of the MCLD during OCV-AST analysis is shown in Figure 4a–d at BOL (0 h), MOL-1 (10 h), MOL-2 (50 h), and EOL (100 h). During the initial investigation of the unused defective CCM at BOL, the entire MCLD was found to cover a geometric area of 757,016 μm². It should be noted that some catalyst still remained over most of the MCLD surface at the outset. As the sample aged, the propagation of defects at the corners of the MCLD appeared to proceed by the development of sharp cracks and degraded areas at its edges. A study by Pestrak et al. showed that the deformation of the electrolyte membrane had a strong influence on the structural changes of the catalyst layers in CCMs [32]. The results observed in our study supported the idea that the deformation of the membrane had a direct influence on the crack formation and areal growth of the catalyst layer defect from MOL to EOL. The image analysis shows that the total defect area (dark triangular area) increased by 5.4% between BOL and MOL-1, 11.25% between MOL-1 and MOL-2, and 1.6% between MOL-2 and EOL. For a better view of the defect cross section, a 3-D diagram of the MCLD is depicted in Figure 4e, while a cross sectional view showing the depth of the defect as one moves along the blue line indicated in Figure 4e is presented in Figure 4f. The width of the defect is measured to be ~426 μm across the blue line. Note that the depth of the defect is not uniform presumably due to the nonuniform removal of the catalyst during the decal transfer step. Figure 4g presents a schematic image of a cross section of the MCLD to provide a better view of the defect in the CCL.

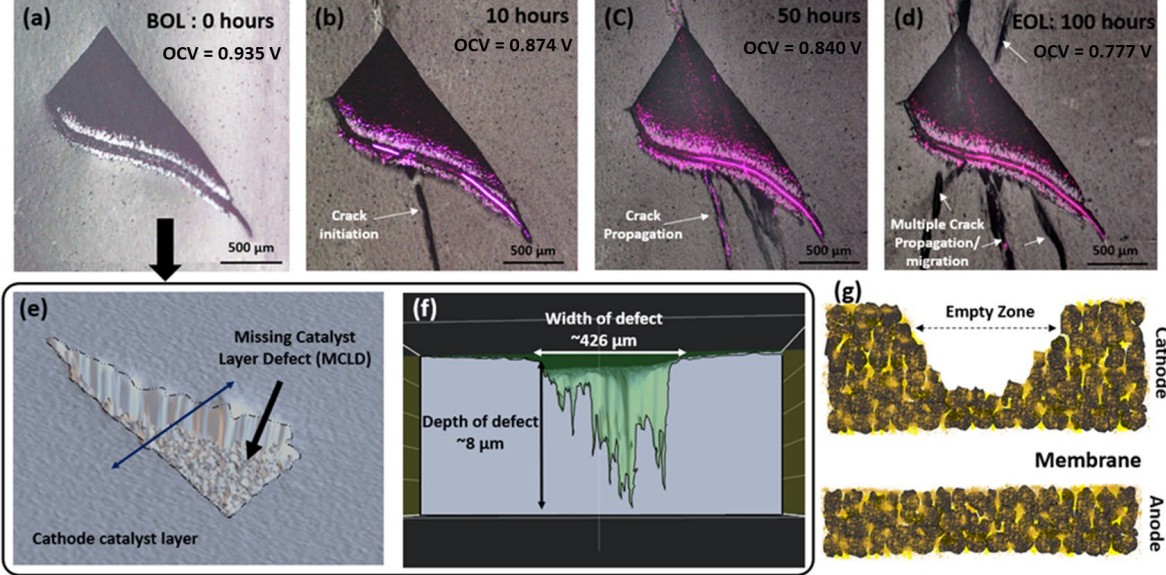

**Figure 4.** Images of defect growth in an MCLD in a defective CCM-1 (**a**) BOL at 0 h, (**b**) MOL-1 after 10 h OCV-AST, (**c**) MOL-2 after 50 h OCV-AST, (**d**) EOL after 100 h OCV-AST, (**e**) 3D graphical view

of MCLD with active area of 757016 μm² on CCL, (**f**) cross-sectional view of MCLD showing dimensions of depth, and (**g**) schematic view of MCLD in CCL across the CCM-1.

After microscopic inspection at BOL, MCLD-CCMs were further examined using IR thermography to assess the criticality of defects such as pinholes, as shown in Steps 1 and 2 in Figure 1. The CCMs that show no evidence of pinholes were then inserted into the custom-designed test cell connected to the fuel cell workstation to resume the monitoring of their OCV. Experiments were conducted on CCMs until their OCV was reduced by ~20% of their initial values. The time evolution of the OCV obtained over the 100 h obtained for a CCM that was defect free (denoted as CCM-baseline) and CCMs that contained an MCLD (denoted as CCM-1-MCLD and CCM-2-MCLD) are included in Figure 5a. CCM-1-MCLD was the same sample shown in Figures 3 and 4 that was removed at MOL in order to track the evolution of the defects using microscopic examination. As shown in Figure 5a, fluctuations of the OCV were observed in the case of CCM-1-MCLD at 10 and 50 h. These times coincided with the stop/start of the OCV-AST to enable the removal and reinstallation of the electrode from the test cell for microscopic examination. However, in order to compare the evolution of the OCV in a CCM that is defect-free with one that contains an MCLD, we used another CCM denoted as CCM-2-MCLD that had a similar MCLD defect to that in CCM-1-MCLD in terms of shape, size, and orientation, but that is never removed for examination until the termination of the OCV-AST. The results of the polarization analysis of both CCMs were discussed in Section 3.4. It should be noted that the main object of the research was to understand the morphological changes of MCLD during aging.

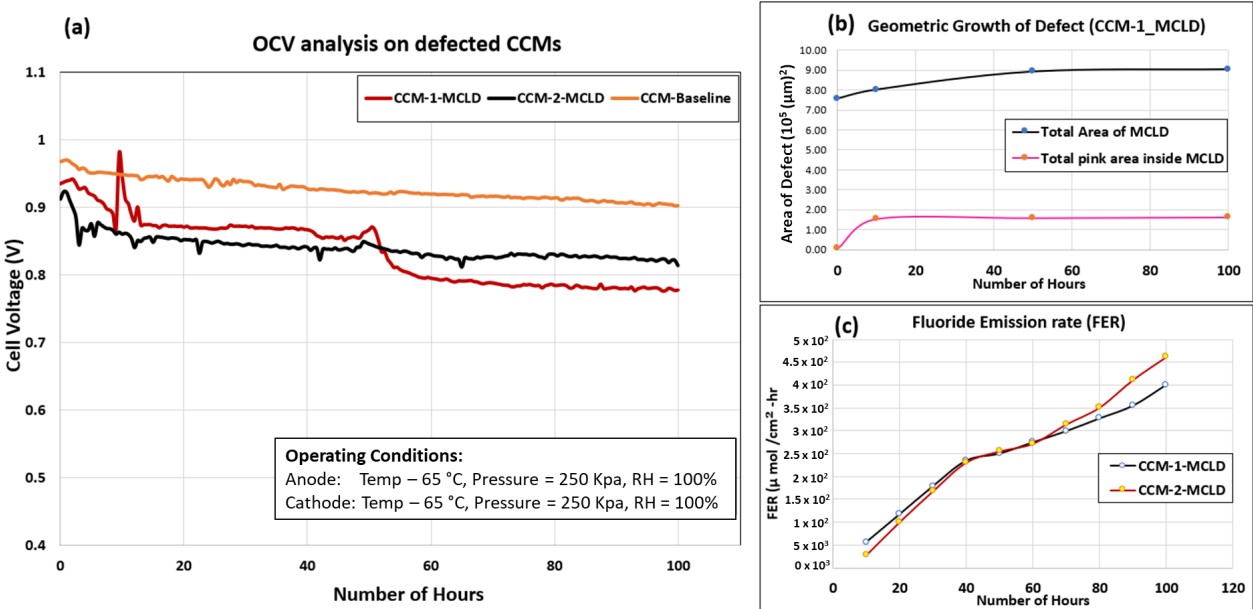

**Figure 5.** (**a**) OCV measurement of defective and nondefective CCMs, (**b**) areal growth of MCLD, and (**c**) fluoride emission rate (FER) at the cathode over the course of 100 h OCV-AST (Figure 4a–d).

Although CCM-2-MCLD and CCM-baseline did not reach their EOL at the end of 100 h, the experiments were terminated at this point since the aim was to compare the rate of OCV degradation with that of CCM-1-MCLD. The OCV of CCM-1-MCLD was found to drop more severely than that of the other two—from 935 mV to 776 mV at an average degradation rate of 1.6 mV/h. The OCV of CCM-2-MCLD degraded from 923 mV to 820 mV at an average rate of 1.0 mV/h. The OCV of the CCM baseline degraded more slowly than the defective CCMs from 968 mV to 909 mV at a rate of 0.6 mV/h. As noted previously, the testing of CCM-1-MCLD was interrupted at time intervals of 10, 50, and 100 h for inspection of the morphological changes in MCLD. As evident in Figure 5a (red line),

this disturbance had some effect on the measured OCV. No such disturbance was observed in the case of CCM-2-MCLD which was removed from the cell for microscopic inspection only at the end of 100 h of OCV measurement. In both cases, similar types of catalyst degradation were observed inside the MCLD, as will be described in Section 3.3.1.

A primary focus of this study was to investigate the relation between the evolution of the MCLD and its loss in OCV as the CCM ages. As shown in Figure 5a (red line), CCM-1-MCLD experienced a noticeable voltage degradation of 7.4% over the first 10 h of operation as its voltage decreased from 935 mV to 865 mV. A particularly noticeable voltage drop of 36 mV from 901 mV to 865 mV between the 9th and 10th hours of OCV was observed. To determine if this loss in performance was reflected in a change in the MCLD, we carried out an image analysis of the CCM after the 10th hour (i.e., MOL-1). Further image analysis was performed on the aged CCM after 50 h (MOL-2) and 100 h (EOL) when the OCV was observed to drop by 10.1% and 17.8%, respectively, with respect to its initial voltage.

Two changes were observed in the first image analysis carried out at MOL-1. First, as shown by comparing Figure 4a,b, crack initiation occurred in the catalyst layer, most likely due to the deformation of the membrane. Second, a significant amount of catalyst had been lost inside the MCLD, as reflected in the significant increase in the pink area in Figure 4b relative to that in Figure 4a. The loss of catalyst particles in the local defect sites stemming from catalyst erosion would significantly facilitate gas crossover of $H_2$ from the negative side to the positive side of the cell, which was one possible cause of the voltage loss at MOL-1.

A comparison of the dark triangular area in Figure 4b with that in Figure 4a shows that the MCLD grows by 5.9% with an aspect ratio of 4.4 over the first 10 h. The cumulative total areal growth of the MCLD (black curve) and the total catalyst loss inside the MCLD (pink area) with respect to time are shown in Figure 5b. Furthermore, the total pink area inside the MCLD increased by more than 2000%, indicating extensive catalyst degradation within the defect.

Over the interval between MOL-1 and MOL-2, the OCV diminished at a rate of 0.85 mV/h from 874 mV to 840 mV. However, minimal growth of the MCLD was observed over this period. Most of the CCM damage appeared to involve the propagation of cracks in the catalyst layer (compare Figure 4b,c). At this stage, the MCLD area (dark triangular area) shown in Figure 4c increased by 11.3%, relative to that measured at MOL-1 with an aspect ratio of 4.11, whereas the growth of the pink area increased by only 2.7% with respect to MOL-1 (Figure 5b). Overall, the catalyst loss in the defect appears to have stabilized between MOL-1 and MOL-2.

Over the last interval from MOL-2 to EOL after 100 h of testing, the OCV dropped from 864 mV to 777 mV corresponding to a degradation rate of 1.74 mV/h. During this period, previously initiated cracks propagated significantly, and new cracks formed on the catalyst layer surface. Figure 3b showed the overview of the damaged catalyst surface with severe cracks at the inlet, center, and outlet locations. According to Figure 4d, no significant change to the MCLD area (dark triangular area) was observed. The pink area within the MCLD increased by only 0.8%, with respect to MOL-2. Numerous new cracks formed, while some merged in the area near the MCLD due to the deformation of the membrane. To conclude, the MCLD was found to grow rapidly during MOL-1, most likely due to chemical degradation that led to catalyst erosion inside the defect (as reflected by the significant growth of the pink area during MOL-1) before the degradation stabilized over the remaining period until the EOL (Figure 5b). It is interesting to observe that the differences in the OCV responses for the defect-free and defective samples followed a similar trend. As evident in Figure 5b, the largest difference in the responses was during the first few hours of the OCV-AST when the presence of the MCLD led to a significant decrease in the OCV. Thereafter, the decline of the OCV followed a similar trajectory in the case of the defect-free and defective CCMs. The largest drop in the OCV coincided with the period when the MCLD was growing most rapidly. Based on these experiments,

the MCLD no longer had a significant influence on the OCV once it had stabilized. Presumably, other events such as crack formation, delamination, catalyst erosion, and membrane pinhole formation contributed more strongly to the longer-term deterioration in cell performance. These observations also suggested that the OCV measurement of a fresh CCM may serve as a useful *in-situ* indicator of the presence of an MCLD generated during MEA fabrication. In the following section, specific regions of the degraded area inside the MCLD were investigated at 100× magnification to more closely examine the changes in defect morphology, and possible mechanisms for degradation of the catalyst layer are discussed. Determination of the precise mechanism for voltage degradation which normally involves the cascade of several events during cell operation was complicated and beyond the scope of the current study.

### *3.3. Degradation Mechanism of Catalyst Layer Defects*

As fuel cells operate, their components break down by chemical and mechanical degradation [33]. Eventually, degradation can lead to the formation of pinholes that terminate the life of the electrode. Previous research has shown that catalyst layer degradation causes gas crossover in the electrode leading to further chemical degradation which in turn causes more cracks or pinholes to form [34–38]. However, no visual evidence of surface degradation and MOL aging mechanisms has been reported in the literature. In this study two types of degradation mechanism were observed in the catalyst layers: (i) surface degradation caused by catalyst erosion and (ii) crack degradation due to membrane deformation.

### 3.3.1. Surface Degradation (Chemical) of Catalyst Layer Defects

To visualize the extent of degradation inside the MCLD defect, we carried out a microscopic image analysis with and without transmitted light. Figure 6a shows the reflected microscope image of the MCLD at MOL-2 obtained without transmitted light. In this image, highly damaged sites in the CL are not completely visible. When this image was captured in transmission mode without reflected light, the damaged areas became more evident. Thin and completely missing catalyst zones and crack propagation are visible in the resulting image in Figure 6b. The white pixels in Figure 6b correspond to the thinnest areas of the catalyst which allow the transmitted light to pass through the defect.

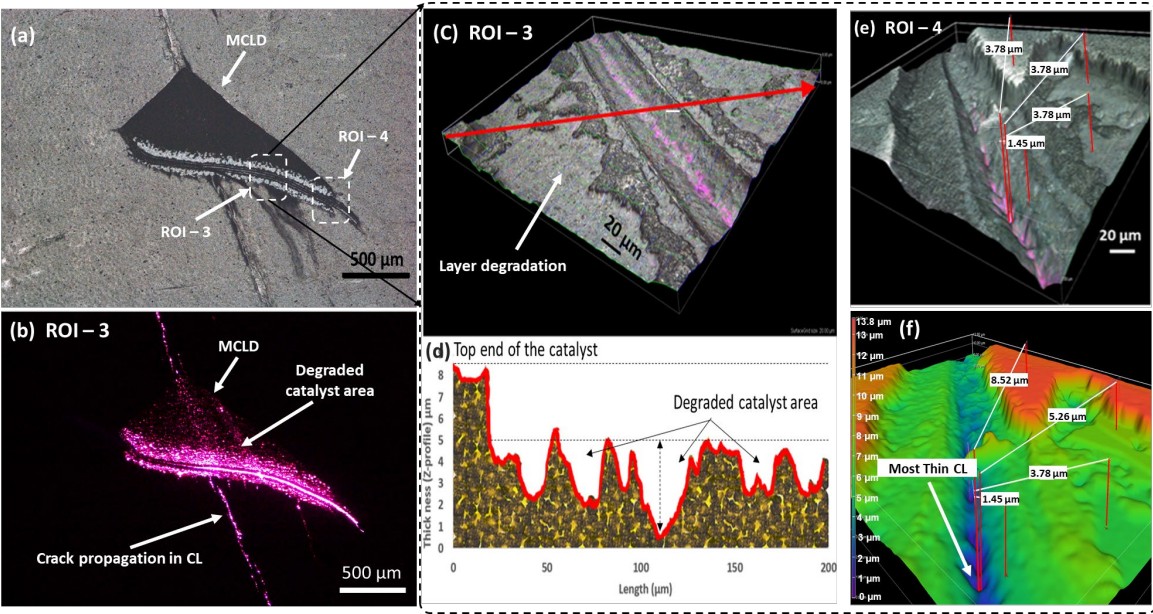

**Figure 6.** (**a**) Reflected microscope image of MCLD at MOL-2 indicating selected regions ROI-3 and ROI-4; (**b**) transmitted light microscope image of MCLD with pink area indicating degraded catalyst

due to chemical degradation during fuel cell operation; (**c**,**e**) are 3D microscopic visualizations of enlarged regions ROI-3 and ROI-4, respectively; (**d**) Z-profile/height scan showing variation in thickness of degraded catalyst layer surface along the red line in Figure 6c; (**f**) colour mapping of morphological features of degraded catalyst layer: blue area represents thinnest catalyst, green represents uneven surface of degraded catalyst layer, orange represents nondegraded catalyst layer.

Regions ROI-3 and ROI-4 located at the middle and edge of the defect (Figure 6a) were selected for closer examination using transmitted and reflected light to characterize surface degradation within the MCLD. The catalyst surface in these two regions had clearly become irregular due to chemical/layer degradation inside the MCLD (Figure 6e,f). Previous studies have shown that these effects can be caused by ionomer leaching from the catalyst layers and $H_2O_2$ (hydrogen peroxide) formation at the defective catalyst sites [33].

Catalyst Erosion

Catalyst erosion is a major problem that lowers the durability of the Pt/C catalyst contained in fuel cell electrodes. As discussed previously in Section 2.2, the ionomer network in the catalyst layer develops strong interfacial bonding with catalyst particles and the electrolyte membrane during hot pressing. This is crucial to enhance the kinetics of the electrochemical reactions and transport of protons. Any nonuniform distribution of ionomer in the catalyst layer and nonuniform thermal compression during hot pressing can lead to the incomplete transfer of the catalyst layer onto the membrane (Figures 2 and 4). We have observed that catalyst particles located within the incomplete transfer zones (MCLD) gradually degrade during operation and are not as stable as catalyst particles in defect-free areas. Examination of CCM electrodes at MOL revealed that weak zones in the catalyst layer were more likely to erode as they were exposed to incoming reaction gases at different relative humidity, pressure, and temperature. For example, Figure 4b obtained at MOL-1 shows that a significant amount of catalyst within the MCLD (pink area indicates the loss of catalyst) had been removed. A potential cause of the catalyst loss is the weak interaction between the Pt/C and ionomer inside the MCLD, as these areas might not have been reinforced at elevated temperatures. As a result, catalyst particles became detached from the thin layer, leading to discontinuities in electronic and ionic flow. Such sites would provide little or no support for electrochemical reactions during fuel cell operation, especially at high current densities.

The effect of catalyst erosion was observed mainly inside the MCLD between BOL and MOL-1 (Figure 4a–d), as determined from the quantitative analysis of the growth of its area (total pink area) over time (Figure 5b). The area of the MCLD increased by ~2000% from its initial value by the time that MOL-1 was reached, but only by 2.7% more between MOL-1 and MOL-2 and 0.8% between MOL-2 and EOL. This trend showed that the weakly bonded catalyst particles at defective sites eroded rapidly in the early stages of operation before quickly stabilizing. This observation suggested that the particles inside the defective area may not be completely impregnated with the catalyst since they did not undergo uniform thermal compression during hot pressing. This causes poor bond strength between adjacent catalyst particles and intact sulphonic acid groups (ionomers) that enabled gradual detachment of the catalyst during cell operation. A complete overview of the degraded area of MCLD is clearly seen in Figure 6b captured under transmitted light mode. The intensity of the light increased across thin areas where the catalyst was lost and the pink area inside the MCLD slowly grew. However, our investigation of various defective CCMs showed this same type of catalyst loss was not exhibited in all the MCLD areas and that this phenomenon depended on defect dimensions, thickness, and location. Although the structural changes at BOL were difficult to estimate, they ultimately affected the integrity of the catalyst layer in the CCM electrode.

### 3.3.2. Mechanical Delamination of Catalyst Layers

Figure 7a shows evidence consistent with catalyst layer delamination having occured at ROI-2. Kai et. al. showed that static and humidity cycling caused deformation of the membrane and extensive crack propagation in the catalyst layer [39]. When a membrane underwent swelling, its pores gained moisture from the reaction gases and became thicker. In this situation, the catalyst layer could have difficulty accommodating the pressure developed by the membrane with the result that surface cracks form. The opposite effect occured during shrinking/drying, i.e., the pores of the electrolyte membrane lost moisture, and the membrane shrank in thickness, causing the catalyst layer cracks to collapse or merge [18]. Ultimately, the combined effects of crack propagation and merging can cause delamination and detachment of the catalyst layer from the membrane.

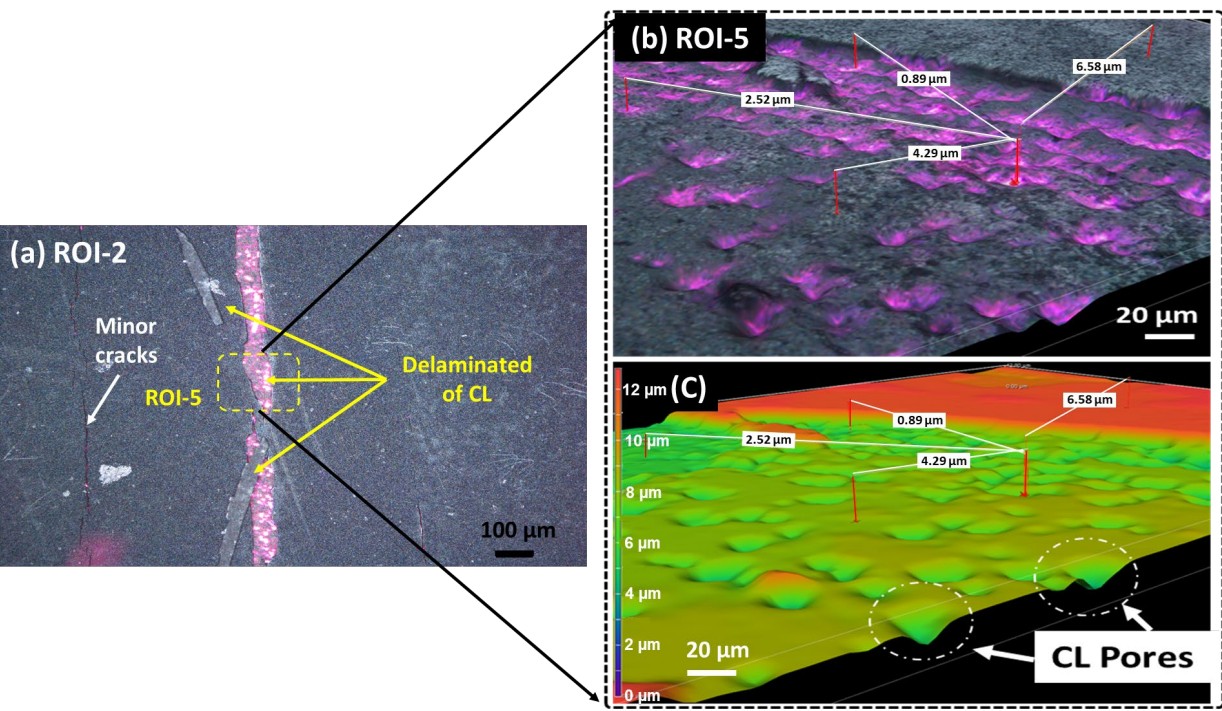

**Figure 7.** Microscopic images of crack propagation on the CCLs at EOL: (**a**) delaminated catalyst layer, (**b**) ROI-5: 3D enlarged view of delaminated and degraded catalyst layer, and (**c**) color mapping of delaminated area of ROI-5 showing catalyst layer pores.

Even after 100 h of OCV-AST operation, it was observed that cracks had not propagated significantly in the vertical direction (i.e., not increased in depth) and had not penetrated beyond ~50–70% of their thickness (Figure 7a). Crack penetration and delamination had affected 50–70% of the catalyst layer. Closer visual examination at 100× magnification was conducted on a specific delaminated region labelled ROI-5 within ROI-2 (Figure 7a). A view of ROI-5 is presented in Figure 7b,c showing that a portion of the catalyst layer detached from the surface crack. Several Z-profile measurements across the delaminated areas on the CCM (inlet, middle, and outlet areas) were carried out to estimate the thickness of delaminated catalyst layers. This analysis revealed that the average thickness of delaminated CLs that originally ranged from 10–15 μm has decreased to 2–6 μm at EOL (some of these regions are shown in Figure 7a). The pink areas in Figure 7b represent the thinnest areas where the transmitted light could pass through the solid catalyst layers. The map showing the height of the catalyst layer in region ROI-5 is presented in Figure 7c. The orange areas represent defect-free catalysts, while the green areas correspond to defective catalyst remaining after delamination. The possible causes of the surface delamination may be traced back to the hot press step during fabrication. The uppermost catalyst surface that is first exposed to the hot plates during CCM fabrication presumably has

a strong and uniform contact bond with Pt/C catalyst particles and ionomer composite in this portion of the catalyst layer [40]. When the lower portions of the catalyst layer became heated, the ionomer network in the catalyst layer also developed strong bonds with the ionomer in the electrolyte membrane [17]. At the interface between the lower catalyst layer and membrane, some portions of the ionomer migrated into the membrane and left behind large pores in the lower catalyst layer. This was also consistent with the idea that CCMs strongly interact with the CL and polymer membrane to reduce the contact resistance between adjacent layers [18]. When the membrane became humidified, it swelled and experienced stress elongation. This set up a pressure difference between the upper and lower portions of the catalyst layer that could lead to delamination in the middle of the catalyst layer. The delamination occured depending on the ionomer network in the catalyst layer. Some of the factors in the network that influenced delamination were catalyst layer thickness, membrane swelling, and ionomer concentration in the catalyst layers. On the cathode, the delamination was observed to leave behind ~30% of the catalyst layer thickness at the least. However, delamination on the anode was rarely observed. An important factor behind the difference in delamination of the two electrodes was that the anode was initially much thinner than the cathode. The white dotted circles in Figure 7c indicated larger pores in the catalyst layer after delamination. To investigate the pore size distribution, image analysis was performed on the defect-free and delaminated areas. The pore diameters were measured to be ~1–2.5 μm in nondefective areas, whereas they were more than five times as large with diameters of ~5–25 μm in the defective areas. Large pores are not favorable for fuel cell operation since they do not contain catalyst and become dead zones for electrochemical reactions and facilitate gas crossover and water flooding since water tends to accumulate at these sites, particularly at high current densities. From our overall examination of CCMs at BOL, MOL-1, MOL-2, and EOL, catalyst erosion is observed only in the defective areas, i.e., inside MCLDs and inside delaminated catalyst layers. No erosion effect was observed in defect-free areas. These observations are particularly important for the development of catalyst coatings and the integrity of catalyst layer fabrication. Although experiments in this study have been done at a particular RH and temperature, future research to investigate crack propagation when the RH varies in cycles would be useful.

### 3.4. Polarization Analysis

Figure 8a shows the polarization curves obtained using defective CCMs containing MCLDs (CCM-1 and CCM-2) and defect-free CCM (baseline) at BOL and EOL. The experiment was carried out at 90 °C with a relative humidity of 80%/80% on the anode/cathode sides by increasing the current density from zero up to a maximum of 2A/cm². The same CCMs examined at BOL are characterized later at EOL after undergoing 100 h of OCV-AST, as discussed in Section 2.2.3. The curves indicate that the polarization of the cell at BOL was not strongly affected by whether or not the CCM contained a defect. However, once the cells were subjected to the OCV-AST, the degradation in performance at EOL was much larger in the case of CCM-1-MCLD (red) compared to CCM-2-MCLD (blue) and CCM-baseline(yellow). In the case of CCM-1-MCLD, a significant drop in polarization curves was observed from CCM-1-MCLD-BOL (black) and CCM-1-MCLD-EOL (red), where its OCV reduced by 11.6% from 982 mV to 868 mV (loss of 114 mV) on going from BOL to EOL. This voltage loss was presumably associated with gas crossover through damaged areas in the electrode. At a current density of 0.1 A/cm², the cell voltage was reduced by 12.1% (102 mV), from 843 mV to 741 mV, after undergoing the OCV-AST. This voltage loss is attributed to reduced electrocatalyst activity. Additional causes for losses in the activation region were catalyst erosion, ionomer leaching, and loss of active catalyst through delamination, as discussed in Section 3.3. Not surprisingly, larger shifts in the voltage and power curves were observed at a higher current density of 2 A/cm², i.e., voltage drops from 415 mV to 270 mV and peak power density decreases from 830 mW to 540 mW in the mass transport region. The observed trend in CCM-1-MCLD in the activation,

ohmic, and mass transfer regions was directly related to defect propagation in aged CCMs.

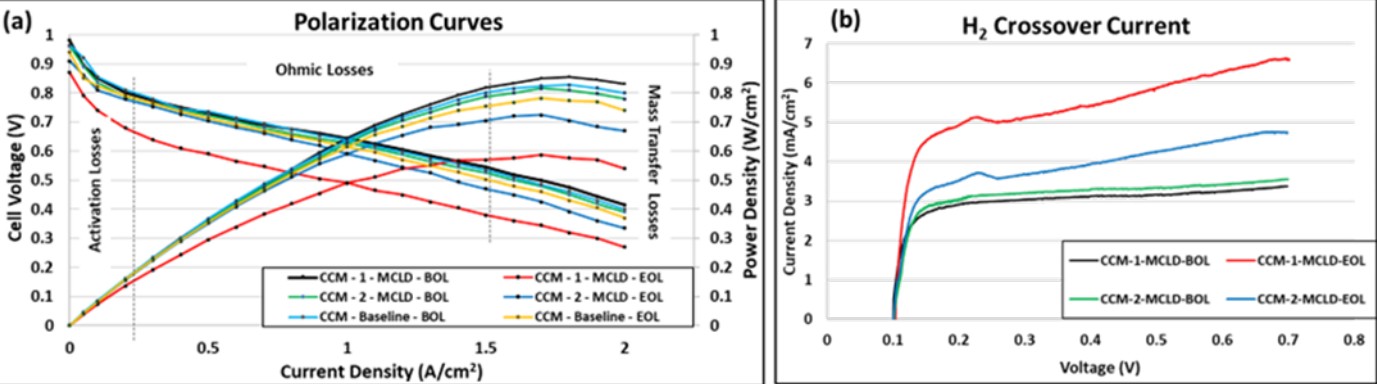

**Figure 8.** (**a**) Polarization curve of CCM-1, CCM-2, and CCM-baseline operated at 90 °C and 80/80% RH on anode and cathode at a pressure of 250 kPa. (**b**) Hydrogen crossover current of defective CCMs (CCM-1 and CCM-2) obtained at a scan rate of 2 mV/sec.

Obviously, another possible cause for some of the voltage loss in CCM-1-MCLD at EOL was its removal and examination twice during the OCV-AST (after 10 and 50 h) to characterize the MCLD using optical microscopy. This could have caused some of the membrane to dry out, leading to structural damage to the catalyst layer, an increase in the ohmic resistance of the electrode, and irreversible voltage loss in the cell during polarization. On the other hand, CCM-2-MCLD was not removed during the OCV-AST, and hence its polarization response was more suitable for comparison with that of the nondefective CCMs. At zero current density, the OCVs measured in CCM-2-MCLD and CCM-baseline decrease by 5.6% (from 0.962 V to 0.908 V) and 2.5% (from 0.966 V to 0.941 V), respectively, between BOL and EOL. At the maximum current density (2 A/cm$^2$), the cell voltage decreased by 9.7% (from 0.389 V (green) to 0.351 V (blue)) in CCM-2-MCLD and 5.2% (from 0.400 V (orange) to 0.379 V (yellow)) in CCM-baseline between BOL and EOL. These results show that the performance deteriorated to a greater extent at both low and high currents when the CCM initially contained an MCLD than when it is defect free. This finding suggests that the crossover flux had a major impact on the cell potential at zero and low current density (activation region), but not at higher currents. Kreitmeier et.al showed that gas permeability across membranes decreases at a high current density which leads to an increase in humidity and membrane swelling and a reduction in the size of pinholes in the membrane [36]. On the other hand, the minimal losses seen in the mass transfer region were associated with water flooding in damaged areas across cracks, delaminated areas, and empty sites within the MCLD. A closer study of the effect of current density on MCLDs would be helpful in gaining a deeper understanding of the influence of catalyst layer defects on cell performance.

### 3.4.1. Hydrogen Crossover

Hydrogen crossover was strongly related to the condition and durability of the fuel cell electrode. Furthermore, its effect was heightened when electrodes were tested under OCV when H$_2$ consumption at the anode was negligible. Under these conditions, H$_2$ slowly diffused through the electrolyte membrane, reacted at the cathode, and thereby reduced the cell potential. The transport of H$_2$ was obviously easier if the catalyst layer and membrane contained defects and pinholes. At the same time, the crossover of gas from the anode to the cathode and vice versa can lead to the formation of H$_2$O$_2$ which further decomposes ionomer in the catalyst layers. The overall health of the electrode at BOL and EOL can be characterized by measuring hydrogen crossover at regular intervals. The black and green curves in Figure 8b show that the crossover current density of

~3mA/cm² measured at BOL was stable for CCM-1 and CCM-2 at voltages above 0.2 V, indicating that the CCM was intact even in the presence of MCLD. Once the CCM was aged for 100 h during the OCV-AST, a significant rise in the crossover current density was observed at all voltages. The blue curve corresponds to CCM-2-MCLD-EOL, while the red curve represents CCM-1-MCLD-EOL in Figure 8b. This significant increase in the crossover current in CCM-1-MCLD-EOL was likely caused by the propagation of cracks and catalyst layer defects that led to pinhole formation in the membrane.

### 3.5. Summary of Defect Analysis

Based on the defect analysis observed from our investigation on defective and nondefective CCMs, a detailed summary of defect growth, loss of OCV, and $H_2$ crossover current are listed in Table 1. To the best of our knowledge, defect investigation at MOL in CCMs is not currently available in the literature. These findings are critical for CCM manufacturers on a commercial scale for the acceptance or rejection of CCMs prior to the fabrication of MEAs. The results also provide valuable insight into quality control methods and the lifetime of fuel cell stacks, which is beneficial to the research community.

**Table 1.** Summary of defective and nondefective CCMs investigated in this study.

|  | **CCM-1** | | | **CCM-2** | | **CCM-Baseline** | |
|---|---|---|---|---|---|---|---|
| Number of MCLD | 1 | | | 1 | | No defects | |
| % Growth of Defect | MOL-1 | MOL-2 | EOL | - | | - | |
|  | 5.9% | 11.2% | 1.3% | | | | |
| OCV (V) | BOL | | EOL | BOL | EOL | BOL | EOL |
|  | 0.937 | | 0.776 | 0.923 | 0.820 | 0.968 | 0.909 |
| H2 Crossover (mA/cm²) | 3.13 | | 5.42 | 3.32 | 3.95 | 3.25 | 3.46 |

### 4. Conclusions

This study has focused on the morphological changes of catalyst layer defects in CCMs (manufacturing defects) under operating conditions and their effects on overall fuel cell performance. The proposed protocol for investigating defects is a nondestructive method that is advantageous in providing areal visualization and failure locations of catalyst layer defects in pristine and aged CCMs. Overall, the effects of chemical and mechanical degradation inside the defective areas are greater and progress more quickly during the initial stages of operation before stabilizing during later stages. The areal dimensions and lost catalyst zones inside the MCLD were measured and examined at regular time intervals and correlated to performance loss. Experiments showed that the voltage degradation rate of defective CCMs was much faster during the initial operating hours until the OCV had dropped by ~10% from its original value. In addition, it was observed that once formed, an MCLD can degrade extensively due to catalyst erosion and ionomer leaching. Examination of the images revealed that mechanical deformation of the membrane leads to 50–70% delamination of the CCL thickness and leaves behind large pores (~ as large as 25μm diameter) on the remaining catalyst layer. In addition, the polarization curves of defective CCMs/MEAs operated at a stack temperature of 90 °C after being subjected to the OCV-AST showed that defect growth had a significant effect on performance. The increase in the hydrogen crossover current was found to be a good indicator of the health of the electrodes as a function of defect growth.

This work is aimed at providing fundamental knowledge on improving the tolerance and durability of CCM electrodes against defects. The microscopic investigation conducted in this study justifies the use of microscopic analysis to visualize the propagation of defects and enables quantitative measurement of the relation between defect growth and projected CCM lifetime. The nondestructive PEMFC electrode investigation approach introduced here should be helpful in improving the quality control of fuel cell technology.

From the observations developed on defect size and defect orientation in CCMs, we also recommend further investigation into the impact of defect location, thickness variation, and empty catalyst sites on the lifetime of fuel cell stacks. Future studies should also focus on the effects of the size of catalyst layer defects in CCMs and membrane damage during fabrication on subsequent MEA performance and/or failure.

**Author Contributions:** Conceptualization, M.P.A., M.F. and M.P.; methodology, M.P.A, M.F. and M.P.; validation, M.P.A. and M.C.; investigation, M.P.A.; writing—original draft preparation, M.P.A.; writing—review and editing, all authors.; visualization, M.P.A.; supervision, M.F. and M.P.; project administration, M.F. and M.P.; funding acquisition, M.F. and M.P. All authors have read and agreed to the published version of the manuscript.

**Funding:** This research work was funded by the Natural Sciences and Engineering Research Council of Canada (NSERC), Grant RGPIN-2020-04149.

**Institutional Review Board Statement:** Not applicable.

**Informed Consent Statement:** Not applicable.

**Data Availability Statement:** Not applicable.

**Acknowledgements:** This work was supported by the Department of Chemical Engineering at the University of Waterloo, Canada Research Chair Tire 1—Zero-Emission Vehicles and Hydrogen Energy systems Grant number: 950-232215, and The Natural Sciences and Engineering Research Council of Canada (NSERC), Discovery Grants Program, RGPIN-2020-04149. The authors thank Dennis Sun, Francine Berretta, and Dr. Sumit Kundu from the Automotive Fuel Cell Cooperation Corporation for their kind support in supplying PEMFC components for the research. The authors also appreciate the contributions of Shankar Raman Dhanushkodi, Brandon Wang, Jonathan Lepine, Kelly Zheng, and Aditi Sharma to the data analysis. Finally, the authors thank Jennifer Moll at the University of Waterloo for technical assistance with reflected microscopy.

**Conflicts of Interest:** The authors declare no conflict of interest exist.

## Abbreviations

| | |
|---|---|
| ACL | anode catalyst layer |
| AST | accelerated stress test |
| BOL | beginning-of-life |
| CCM | catalyst-coated membrane |
| CCL | cathode catalyst layer |
| DOE | Department of Energy |
| EOL | end-of-life |
| FER | fluoride emission rate |
| GDE | gas diffusion electrode |
| GDL | gas diffusion layer |
| LSV | linear sweep voltammetry |
| MCLD | missing catalyst layer defect |
| MEA | membrane electrode assembly |
| MOL | middle-of-life |
| OCV | open-circuit voltage |
| PEMFC | polymer electrolyte membrane fuel cell |
| RH | relative humidity |
| ROI | region of interest |

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
