# Peer review of "Morphological Characteristics of Catalyst Layer Defects in Catalyst-Coated Membranes in PEM Fuel Cells"

_2673-3293, doi:10.3390/electrochem4010001_

Round 1

Reviewer 1 Report

The authors present a novel study on catalyst defects that is of interest to the research community and industry, hence it is worth publishing. 

Some clarifications may be provided as follows:

Line 131: the authors claim the transfer ratios were 97% (±2%) but there is no uncertainty analysis that can back up the claim. How many times were the experiments repeated? What was the precision of the equipment? I suggest providing an uncertainty analysis to back up the claims.

Line 309: Authors mention an imagine that is not included. Why did the authors choose to mention it if it's not included. Perhaps the image can be included to justify the claim.

Figure 5 presents the OCV measurement of defective and non-defective CCM. The slope is relatively similar for all three cases however, jumps are observed at 10h and 50h for CCM-1, which is expected given that the sample is inspected at those times. Can the authors explain whet happened at 50h to cause such a significant drop in the OCV? The sample was inspected at 10h and a significant drop was not observed - how come? Are the results reproducible?

Author Response

Dear Reviewer, 

We would like to thank you for providing valuable comments and suggestions, which have strengthened the quality of this paper. Please see the attachment for a comprehensive response to your comments and suggestions. 

Thank you very much. 

Reviewer 2 Report

Dear Authors,

Thank you for submitting your work to Electrochem. The overall quality of the article is good. There are several minor things can make the paper better.

1) Add a table of your abbreviations.

2) Add a summary table of all important data for CCM -1, CCM-2 and CCM -baseline at BOL, MOL and EOL.

3) Add the hydrogen crossover curve of CCM-2 as well in Figure 8 b) as well. 

Best, 

Author Response

Dear Reviewer, 

We would like to thank you for providing valuable comments and suggestions, which have strengthened the quality of this paper. Please see the attachment in response to your comments and suggestions. 

Thank you very much.  
